# Shade and Altitude Implications on the Physical and Chemical Attributes of Green Coffee Beans from Gorongosa Mountain, Mozambique

Crimildo T. Cassamo [1,2,3,†], Adilson V. J. Mangueze [1,4,†], António E. Leitão [1,4,†], Isabel P. Pais [4,5,†], Rita Moreira [5], Claudine Campa [6], Rogério Chiulele [7], Fabrício O. Reis [8], Isabel Marques [1], Paula Scotti-Campos [4,5], Fernando C. Lidon [4], Fábio L. Partelli [9], Ana I. Ribeiro-Barros [1,4,*] and José C. Ramalho [1,4,*]

1 PlantStress & Biodiversity Lab, Centro de Estudos Florestais (CEF), Instituto Superior Agronomia (ISA), Universidade de Lisboa (ULisboa), Quinta do Marquês, Av. República, 1349-017 Lisboa, Portugal
2 Nova School of Business and Economics, Universidade NOVA de Lisboa (NOVASBE), 2775-405 Carcavelos, Portugal
3 Faculdade de Engenharias e Tecnologias, Campus da Lhanguene, Universidade Pedagógica de Maputo, Av. do Trabalho, Maputo 248, Mozambique
4 Unidade de Geobiociências, Geoengenharias e Geotecnologias (GeoBioTec), Faculdade de Ciências e Tecnologia (FCT), Universidade NOVA de Lisboa (UNL), Monte de Caparica, 2829-516 Caparica, Portugal
5 Unidade de Investigação em Biotecnologia e Recursos Genéticos, Instituto Nacional de Investigação Agrária e Veterinária, I.P. (INIAV), Quinta do Marquês, Av. República, 2784-505 Oeiras, Portugal
6 Institut de Recherche pour le Développement (IRD), Diversity-Adaptation-Development of Plants (UMR DIADE), 911 Avenue Agropolis, 34394 Montpellier, France
7 Department Produção e Protecção Vegetal (PPV), Faculdade de Agronomia e Eng. Florestal, Universidade Eduardo Mondlane, Av. Julius Nyerere/Campus, Maputo 3453, Mozambique
8 Centro de Educação, Ciências Exatas e Naturais (CECEN), Department de Biologia (DBio), Universidade Estadual do Maranhão (UEMA), Av. Lourenço Vieira da Silva, 1000, Cidade Universitária Paulo VI, Jardim São Cristóvão, São Luís 65055-310, MA, Brazil
9 Centro Universitário do. Norte do Espírito Santo (CEUNES), Department Ciências Agrárias e Biológicas (DCAB), Universidade Federal Espírito Santo (UFES), Rodovia BR 101 Norte, Km. 60, Bairro Litorâneo, São Mateus 29932-540, ES, Brazil
* Correspondence: aribeiro@isa.ulisboa.pt (A.I.R.-B.); cochichor@isa.ulisboa.pt or cochichor@mail.telepac.pt (J.C.R.)
† These authors contributed equally to this work.

**Abstract:** *Coffea arabica* L. is as a tropical crop that can be grown under monocrop or agroforestry (AFS) systems, usually at altitudes greater than 600 m, with suitable environmental conditions to bean quality. This study aimed to assess the effect of altitude (650, 825, and 935 m) and light conditions (deep shade—DS, and moderate shade—MS provided by native trees, and full Sun—FS) on the physical and chemical attributes of green coffee beans produced in the Gorongosa Mountain. Regardless of altitude, light conditions (mainly MS and FS) scarcely affected most of the studied physical and chemical attributes. Among the few exceptions in physical attributes, bean mass tended to lower values under FS in all three altitudes, whereas bean density increased under FS at 650 m. As regards the chemical compound contents, sporadic changes were found. The rises in trigonelline (MS and FS at 935 m), soluble sugars (FS at 935 m), and the decline in *p*-coumaric acid (MS and FS at 825 m), may indicate an improved sensory profile, but the rise in FQAs (FS at 825 m) could have a negative impact. These results highlight a relevant uncertainty of the quality changes of the obtained bean. Altitude (from 650 to 935 m) extended the fruit maturation period by four weeks, and altered a larger number of bean attributes. Among physical traits, the average sieve (consistent tendency), bean commercial homogeneity, mass, and density increased at 935 m, whereas the bean became less yellowish and brighter at 825 and 935 m (b*, C* colour attributes), pointing to good bean trade quality, usually as compared with beans from 650 m. Furthermore, at 935 m trigonelline and 5-CQA (MS and FS) increased, whereas FQAs and diCQAs isomers declined (in all light conditions). Altogether, these changes likely contributed to improve the sensory cup quality. Caffeine and *p*-coumaric acid showed mostly inconsistent variations. Overall, light conditions (FS, MS, or DS) did not greatly and

consistently altered bean physical and chemical attributes, whereas altitude (likely associated with lower temperature, greater water availability (rainfall/fog), and extended maturation period) was a major driver for bean changes and improved quality.

**Keywords:** agroforestry system; altitude; chlorogenic acids; coffee; colour parameters; full sun; green bean; quality; shade; soluble sugars

## 1. Introduction

To date, 130 coffee species have been identified [1], but only two are responsible for world coffee trade: *Coffea arabica* L. and *Coffea canephora* Pierre ex A. Froehner that contribute with *ca.* 60–65% and 35–40% of global production, respectively [2–4]. World coffee annual production has recently reached *ca.* 10 million tons, generating an income of *ca.* USD 200.0000 million [5], and supporting the livelihoods of about 25 millions of smallholder farmers in the tropical region [6–10] who account for *ca.* 60% of the coffee farms [11].

Among the two main producing coffee species, *C. arabica* can be grown under monocrop or under agroforestry (AFS), and intercropped systems, at altitudes usually between 600 and 2500 m [12,13]. This species have been pointed as quite sensitive to climate changes (CC) and global warming, what could lead to important losses of suitable cropping areas, lower bean yield and quality, and increase pest incidence [14–16]. Importantly, this quality reduction occurs in a context where consumer awareness has increased the demand for better product quality [17]. Despite such sensitivity to CC, recent reports highlighted a greater intrinsic resilience to abiotic stresses than earlier assumed of some elite cultivars [8,18,19]. Furthermore, the "C-fertilization" effect, promoted by the rising air [$CO_2$], can benefit photosynthetic performance and coffee growth [8,20], and mitigate the negative impacts of supra-optimal temperatures and drought [21–23], while preserving coffee bean production and quality [24,25]. Still, it is expected that CC can have some degree of deleterious impacts in the coffee crop, especially in marginal areas/regions. In fact, adverse thermal and water availability can greatly reduce potential yields [26], and because water supply might affect coffee bean quality *per se* [27], it is anticipated a negative interaction between limited water supply and warming, ultimately exacerbating the impact on coffee bean quality. This highlights the urgent need to implement mitigation management practices, such as the use of shade with tree species in Agroforestry Systems (AFS), or intercropping associations. These promote a more efficient use of water, reduce warming at the plant level, and avoid large drops of night temperatures at high elevations, or at lower latitudes, thus reducing chilling and frost damage. Overall, a more suitable microenvironment, concerning both air temperature and humidity, will be obtained for the plants [11,28–30].

Green coffee beans are known to contain, among others, trigonelline, chlorogenic acids (CGAs), and caffeine, which are important quality bean constituents for the coffee industry [31,32]. CGAs isomers are naturally occurring bioactive compounds, with quite known antioxidant activity against free radicals and metal ions, which accumulate in the bean as the coffee fruit matures, which greatly contribute to the final acidity, aroma, flavour, bitterness, and astringency of the coffee beverage [33–37]. In green coffee beans this important group of non-volatile phenolic molecules includes the main subgroups of monocaffeoylquinic acids (CQA), feruloylquinic acids (FQA), dicaffeoylquinic acids (diCQA) and, in smaller amounts, *p*-coumaroylquinic acids. Coffee is also a rich source of another major class of phenolic compounds, the hydroxycinnamic acids that include caffeic, ferulic, and *p*-coumaric acids (*p*-CQA) [35,38–41].

The coffee bean quality is largely associated with their physical and chemical attributes. In turn, these are closely dependent on the coffee species/genotype, microclimatic (temperature, water availability, irradiance) conditions along the fruit/bean maturation process, soil characteristics, as well as of agricultural crop management practices (e.g., fertilization, irrigation), and post-harvest processing, with prevalent environmental conditions during fruit development playing a crucial role in the final bean quality [24,26,42–47]. Among the environmental conditions, those related with shade and altitude are often pointed to have beneficial effects on coffee bean quality, and might be key factors to produce specialty or high-quality coffee in the future. Still, there is some controversy regarding the potential positive impact on bean quality of high altitudes and shading, with AFS being associated either with lower yields and unaffected quality, or improved bean quality [11]. Furthermore, the reduction of irradiance at leaf level caused by shade greatly depends on the density and species of the shading trees. Higher altitudes and shade at lower altitudes might improve the physical and chemical traits of beans, associated with lower temperatures and higher air humidity [28,46,48–50]. Such cooler weather conditions can prolong the coffee fruit/and bean maturation period, promoting greater accumulation of sugar and aroma-related phenolic compounds [6,43].

Although AFS practices have been proposed as a nature-based strategy for coffee farmers to mitigate and adapt to future climate conditions [11], there are somewhat contradictory reports regarding the potential benefits of shade on coffee productivity and quality. In fact, shade was reported to reduce [47], increase [51], or have no impact [52] in coffee yields. Also, shade can alter green bean size and biochemical compounds [43]. However that was found to have negative influence on sensory attributes [53], or to improve bean quality [51]. Finally, other studies point that irradiance/shade by itself seems to have little effect on organoleptic bean quality if nitrogen is not limiting to the plant, revealing that crop management practices can be a greater determinant factor to quality than the genetic factors [54]. Much of these apparently contradicting findings generate controversy about the use of AFS, due to the possibility of negative impacts on coffee growth, yield, and the potential to exacerbate biotic stressors [53], what is also greatly dependent on local environment, shade level/density type of shading trees, and other crop management practices [11,30].

The effect of altitude on the chemical composition of coffee beans, disease incidence, etc., is also in some aspects divergent. For instance, altitude positively influenced the physical attributes of the bean [53], and with higher yield, better quality and fewer defects were reported [50]. However, lower contents of phenolic compounds associated with aroma and bitterness were reported with increased altitude [55], while others studies pointed to opposite findings [56]. Also, high chlorogenic acids contents (CGA) have been associated with either low [38] or high [37] quality coffee. This uncertainty is further exacerbated by the fact that altitude has been studied alone without no great impact [55], or increasing [56] bean acidity, or together with AFS with results pointing to greater cup acidity at greater altitude [55].

Finally, most of the apparent contradictory findings would result from the fact that studies used distinct *C. arabica* cultivars that intrinsically respond differently to shade/altitude, from yield to quality aspects [11,30], and other management conditions that can strongly determine the final outcome [54].

In our case, in addition to the previously mentioned advantages of AFS, the use of this strategy to produce coffee in the Gorongosa Mountain, greatly contributes to the restoration of the degraded rainforest, and its associated biodiversity within the Gorongosa National Park (GNP), while contributes to increase the income and livelihoods of the local farming communities. Located in the center of Mozambique, GNP is one of the most biodiverse places worldwide, with an outstanding history of wildlife restoration.

In this context, the Coffee-AFS system with native trees is being implemented in degraded rainforest landscape, above of 700 m (protected area of GNP), but it is essential to find a combination of adequate light condition (shade density under AFS) and/or altitude that provide sustainable productivity and better bean quality. Therefore, with the present work we analyse important physical and chemical attributes associated with coffee green bean (and, thereafter, cup) quality, from cropped plants in GNP, at three altitudes and submitted to three light irradiance levels. Having in mind coffee bean quality traits, we aim at to evaluate and identify the best environment combinations to implement in this new cropping area in the in GNP area in the Gorongosa Mountain in Mozambique.

## 2. Materials and Methods

### 2.1. Experiment Design

Four year-old coffee plants (*Coffea arabica* cv. Costa Rica 95, from Catimor group) cropped at the Gorongosa Mountain, part of the Gorongosa National Park, Mozambique (Lat. 18°14′ and 18°42′ South and Long. 33°50′ and 33°13′ East), were used to study the impact of light conditions and altitude on the bean physical and chemical parameters associated with quality. The plants were implanted 1.5 m apart within the row and 3 m between rows, for an approximate density of *ca.* 2222 plants ha$^{-1}$, similarly to all light treatments.

To assess the impact of altitude and/or light conditions on the traits of green coffee beans, the experiments were established with a split-plot design, with each altitude representing the main plot and the three assigned levels of light as sub-plot. For each of the three light conditions within each altitude, five plants were randomly selected to provide the green beans to be analysed. For altitude it was considered three levels: 650 (18°30′53″ S, 34°3′5″ E), 825 (18°30′4″ S, 34°2′58″ E), and 935 m above sea level (a.s.l.) (18°28′54″ S, 34°2′43″ E). Within each altitude, the irradiance studies comprised deep shade (DS), moderate shade (MS), and full Sun (FS). These conditions corresponded to an average diurnal PPFD of $127 \pm 28$ μmol m$^{-2}$ s$^{-1}$ (DS), and $725 \pm 48$ μmol m$^{-2}$ s$^{-1}$ (MS), and $1268 \pm 52$ μmol m$^{-2}$ s$^{-1}$ (FS), thus with the PPFD of DS and MS representing *ca.* 10% and 57% of that observed in FS. These diurnal values were taken in clear sky days, in the upper third (illuminated) part of the plant, using the light sensor from an infrared gas analyzer (IRGA) system, model LI-6400 (LI-COR Biosciences, Lincoln, NE, USA). The given values correspond to the average of six months (April, May, June, July and September) along 2020, using the PPFD measurements obtained in four diurnal times per day (9–10 h; 11–12 h; 13–14 h; 15–16 h, with 10 values per hour, 40 values per day, and a total of 200 values for the period from April to September).

Shade was provided by native forest trees, namely *Khaya anthotheca* (Welw.) C. DC., *Erythrina lysistemon* Hutch., *Albizia adianthifolia* (Schumach.) W. Wight., *Bridelia micrantha* (Hochst.) Baill., *Annona senegalensis* Pers., and *Bridelia micrantha* (Hochst.) Baill, exposing the underneath coffee plants to the mentioned irradiation values.

As the system was rainfed, the water availability was obtained by monitoring precipitation through pluviometers installed in the three altitudes (Figure 1). Additionally, air temperature values were monitored with Temperature External Data Logger devices (ONSET, Pro v2 Logger U23-00x, Onset Computer Corporation, MA, USA), installed in each of the three altitudes and three light conditions, collecting values every 15 min along the day. These 96 values per day were used to obtain daily means. Monthly averages in each date were then calculated using the 30 daily average values (Figure 2).

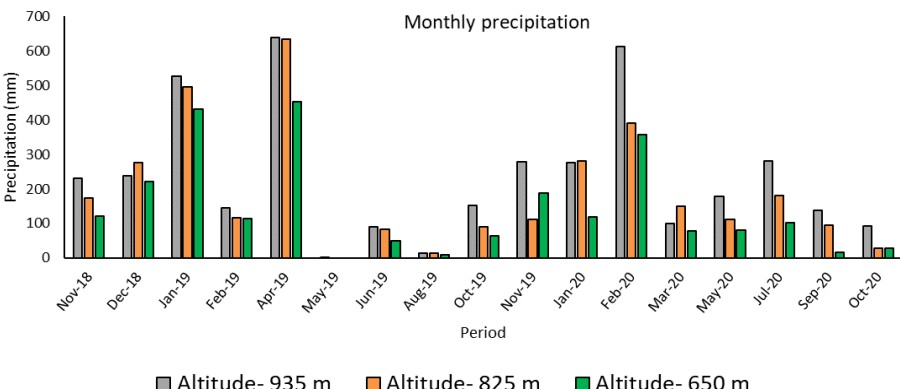

**Figure 1.** Monthly precipitation rainfall values obtained from pluviometers installed at three altitudes (935 m, 825 m, 650 m) in the experimental area of Gorongosa Mountain.

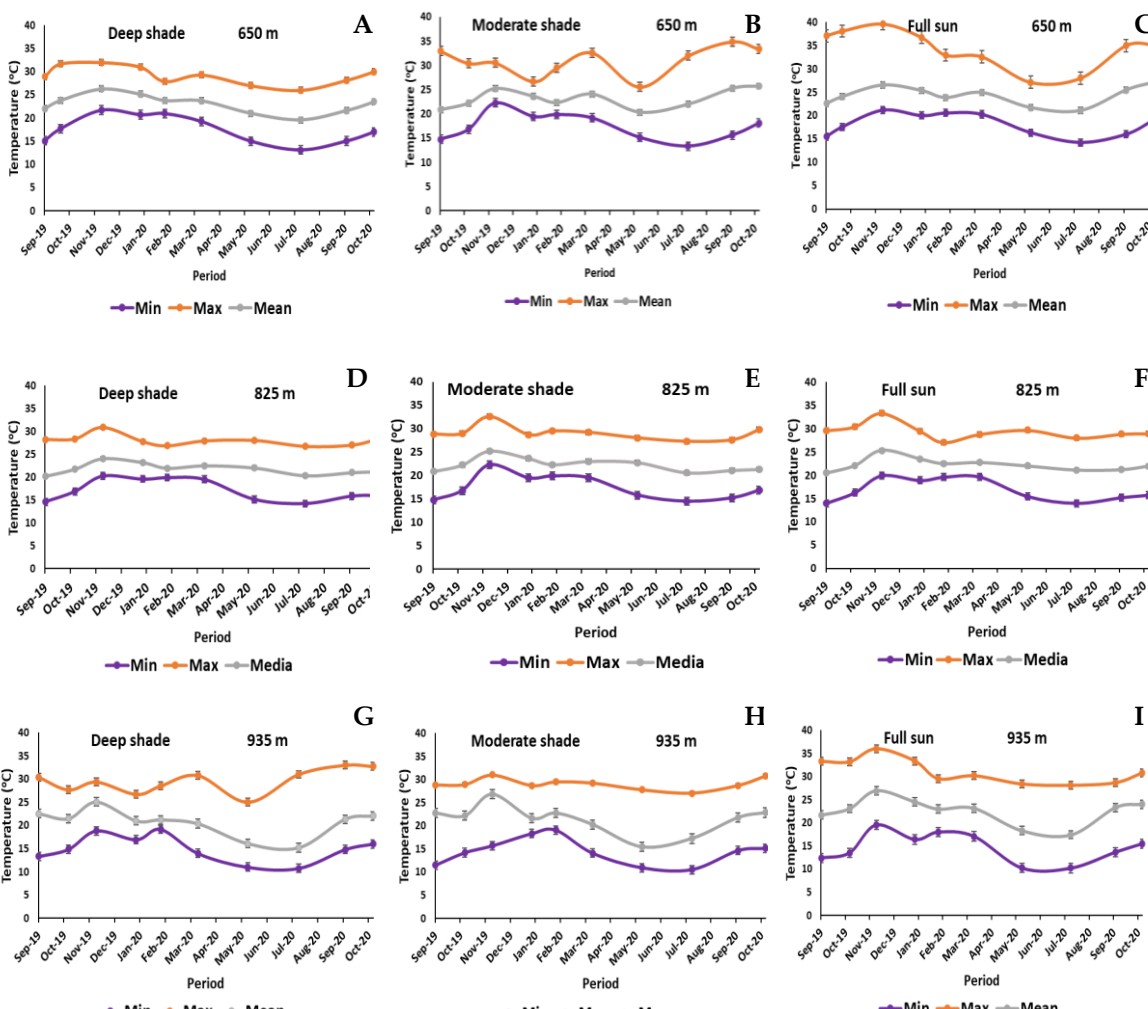

**Figure 2.** Maximum, mean and minimum air temperatures measured, under three light exposure conditions (deep shade, moderate shade and full Sun) and three altitudes (650 m, 825 m and 935 m) in the experimental area of Gorongosa Mountain, obtained with temperature sensors (ONSET, Pro v2 Logger U23-00x), installed in each of the three altitudes and three light conditions. Each point represent the mean ± SE (n = 30).

### 2.2. Fruit Post-Harvest Processing

At full maturation stage (red cherries), approximately 1 kg of fruits were harvested from each of the five plants in each treatment (three altitudes and three light conditions). After manual fruit depulping, the obtained coffee beans were placed in water for 24 h for slight fermentation and mucilage removal. Seeds were then sundried until a moisture content of *ca.* 11–12% was reached, as evaluated by a moisture meter (TG Pro, Coffee and Cocoa, Draminski, Poland). Finally, the parchment was removed, and the few defective coffee beans (broken, atypical colour and/or form) were discarded in each sample according to ISO (2005), before physical and chemical analyses.

### 2.3. Physical Attributes of Green Coffee Beans

Coffee beans with *ca.* 11–12% of moisture were used for this analysis.

#### 2.3.1. Bean Mass of 100 Beans, Apparent Density and Size

Samples of 100 beans per plant were taken and weighed to obtain their mass [57,58].

The apparent density was determined as the ratio between the weight and the volume occupied by the coffee beans [57–59].

Bean size was assessed based on ISO 4150 [60], using several sieves with gradual decreasing diameter: number 20 (8.00 ± 0.19 mm), 19 (7.50 ± 0.18 mm), 18 (7.10 ± 0.18 mm), 17 (6.70 ± 0.17 mm), 16 (6.30 ± 0.17 mm), 15 (6.00 ± 0.16 mm), 14 (5.60 ± 0.16 mm), 13 (5.00 ± 0.15 mm), 12 (4.75 ± 0.14 mm), and 10 (4.00 ± 0.13) [61]. Samples of 100 g of each plant were weighted in portion retained per sieves, and the weight retained in each sieve was converted to the percentage of 100 g of beans.

The commercial homogeneity was evaluated by the sum of the percentages from the two most represented sieves.

#### 2.3.2. Colour Analysis

Determination of colour attributes of ground coffee bean globally followed the procedures previously reported [24,58]. Briefly, colour parameters, lightness (L), and chromaticity (coordinates a* and b*) were obtained with a Minolta CR 400 colourimeter (Minolta Corp., Ramsey, NJ, United States) coupled with a glass container for solid samples (CR-A504). Measurements were performed for illuminant D65 based on Commission Internationale de l'Éclairage (CIE) L* a* b* system. The colourimeter was first calibrated to white Yxy coordinates (Y = 93.10, x = 0.3161, y = 0.3326). L* measures the lightness of a colour and ranges from black (0) to white (100); a* indicates the contribution of red or green (when its value is positive or negative, respectively); and b* the contribution of blue or yellow (when its value is negative or positive, respectively). The elements of perceived colour lightness (L*), Hue angle (H°), chroma (C, saturation), and CI were determinate from the L* a* b* coordinates. Calculation of H° (=(arctg (b*/a*)), in degrees, sets the colour (starting at red, 0°, yellow, up to 90°, green, up to 180°, and blue, up to 270°). C* (=(a*² + b*²)^{1/2}) is a measure of colour saturation or purity, were calculated following [62], whereas CI (=(1000a*)/(L* b*)) [63] ranges between −20 (green) and +20 (orange) with 0 representing yellow.

### 2.4. Chemical Characterization of Green Coffee Beans

To express the data in dry weight, the moisture content of the coffee bean samples was determined according to AOAC [64] procedures referred in [65]. For each plant, 100 green coffee beans were selected, weighed, and dried in an oven at 105 °C for 24 h. After cooling in a room temperature (*ca.* 20 °C), the samples were weighed and the percentage of moisture was calculated.

Chlorogenic Acids, P-Coumaric acid, Caffeine, Trigonelline and Soluble Sugars in the Green Coffee Bean

For chemical analysis, green beans were grounded (<500 μm size), using a grinder (Pulverisette 14, Fritsch, Germany), vacuum packed, and stored at 4 °C until analysis.

Monocaffeoylquinic acids (3-CQA, 4-CQA, 5-CQA), and *p*-coumaric acid, as well as trigonelline and caffeine were analyzed based on [66]. In detail, an aliquot of 500 mg ground beans from each sample/plant was extracted with 30 mL of acetonitrile/water (5:95, *v/v*) for 10 min, at 80 °C, and filtered through Whatman Paper No. 1. A 5 mL aliquot was then diluted to a total volume of 25 mL with the same acetonitrile/water extracting solution, and filtered (nylon, 0.45 μm). A 20 μL aliquot was then injected into an HPLC Beckman System Gold (USA) equipped with a DAD (model 168), and a solvent module (model 126).

For separation a reversed-phase HPLC analysis was performed using an end-capped, C18, 5 μm Spherisorb ODS2 column (250 × 4.6 mm) (Waters, Milford, MA, USA), and a mixture of acetic acid/water (5:95, *v/v*) as eluent A, and acetonitrile as eluent B, at a flow rate of 1 mL min$^{-1}$, at *ca.* 25 °C. A gradient program was used: 5% B during 5 min; from 5 to 13% B during 5 min; 13% B during 35 min, and returned to initial conditions after 1 min. The trigonelline and caffeine compounds detection was performed at $Abs_{272nm}$, whereas for *p*-coumaric, 3-CQA, 4-CQA, and 5-CQA acids the $Abs_{320nm}$ was used.

Additionally, the chlorogenic acids 3-FQA, 4-FQA, 5-FQA, 3,4-diCQA, 3,5-diCQA and 4,5-diCQA were analysed as previously described [31,67]. Samples of 1 g of ground green bean per plant were added to 40 mL of a methanol-water (40:60) solution, stirred for 1 h, and centrifuged (9400× *g*, 5 min, 20 °C), and the supernatant decanted. For samples clearing were added 1 mL of Carrez solution I (aqueous solution of zinc acetate dihydrate and glacial acetic acid, 10.95 g and 1.5 mL, respectively, to a final volume of 50 mL), 1 mL of Carrez solution II (aqueous solution of 5.3 g of potassium hexacyanoferrate II trihydrate in a final volume of 50 mL), and a methanol-water (40:60, *v/v*) solution to obtain the final volume of 100 mL. After 15 min, the mixture was filtered using the Whatman paper filter n° 1, and an aliquot of 10 mL was filtered (nylon, 0.45 μm). The samples were then analysed using the same HPLC system and column used for chlorogenic acids. The elution of a 20 μL injection was performed at *ca.* 25 °C, over 45 min, with a 1 mL min$^{-1}$ flow rate, using an optimized linear gradient 20–70% of B (solvent A tripotassium citrate buffer solution 10 mM, pH 2.5; solvent B methanol 100%). Detection of 4-FQA, 5-FQA, 3,4-diCQA, 3,5-diCQA, and 4,5-diCQA were performed at $Abs_{330\,nm}$.

The identification and quantification of all CQAs and FQAs were carried out using 5-CQA as a standard, following the equation proposed by [68] whereas standard curves were used for caffeine, trigonelline and *p*-coumaric acid, as previously described [24].

Soluble sugars were evaluated in *ca.* 400 mg of ground green coffee beans, after extraction according to [69], following [70]. Briefly, sample aliquots of 40 μL were injected in a HPLC system (Waters, Milford, MA, USA), coupled to a refractometric detector (Waters 2414), and the separation of sugars was performed using a Sugar-Pak 1 column (300 × 6.5 mm; Waters) at 90 °C, with $H_2O$ as eluent (containing 50 mg EDTA-Ca $L^{-1}$ $H_2O$) and a flow rate of 0.5 mL min$^{-1}$. Sugars were identified and quantified, using the individual standard curves for sucrose, glucose fructose and arabinose.

## 2.5. Statistical Analysis

Coffee bean physical and chemical parameters data were analysed using a two-way ANOVA, considering the effects altitude and/or light conditions taking into account the implemented split-plot design. Mean comparison was then performed through a Tukey's HSD test with 95% confidence level, using the software R studio v3.1.

## 3. Results

### 3.1. Altitude and Light and Their Interactions Impact on Green Bean Traits

The results of from variance analysis showed that altitude had the wider and transversal impact among the studied characteristics of green coffee bean (Table 1).

**Table 1.** ANOVA results for altitude and light condition effects and their interaction on the attributes mass of 100 green coffee beans, and bean apparent density, colour lightness (L*), the contributions of red or green (a*), and blue or yellow (b*), as well as the calculated parameters of Chroma (C*), Hue angle (H°), and colour index (CI), trigonelline, caffeine, *p*-coumaric acid, monocaffeoylquinic acids (3-CQA, 4-CQA, 5-CQA), feruloylquinic acids (4-FQA, 5-FQA), and dicaffeoylquinic acids (3,4-diCQA, 3,5-diCQA, 4,5-diCQA), and soluble sugars (sucrose, glucose, fructose, arabinose, total sugars), obtained of green coffee beans obtained under three light exposure conditions (deep shade, DS; moderate shade, MS; full Sun, FS) and three altitudes (650 m, 825 m and 935 m).

| Attributes | Altitude (A) | Light Condition (L) | A * L Interaction | Altitude 650 m | Altitude 825 m | Altitude 935 m | Light Condition DS | Light Condition MS | Light Condition FS |
|---|---|---|---|---|---|---|---|---|---|
| Mass of 100 beans | $<2 \times 10^{-16}$ *** | $1.8 \times 10^{-5}$ *** | ns | ns | ns | ns | ns | ns | ns |
| Apparent density | $2 \times 10^{-6}$ *** | 0.0354 * | ns | ns | ns | ns | ns | ns | ns |
| Commercial homogeneity | $1.6 \times 10^{-5}$ *** | ns | ns | 0.0083 ** | ns | ns | 0.0001 *** | 0.0003 *** | ns |
| Average sieve | 0.0472 * | ns | ns | ns | ns | ns | ns | ns | ns |
| Frequent sieve | ns | ns | ns | ns | ns | ns | ns | ns | ns |
| L* | 0.02319 * | ns | ns | ns | ns | ns | ns | ns | ns |
| a* | ns | ns | ns | ns | ns | ns | ns | ns | ns |
| b* | $4 \times 10^{-6}$ *** | ns | ns | ns | ns | ns | ns | ns | ns |
| C* | $4 \times 10^{-6}$ *** | ns | ns | ns | ns | ns | ns | ns | ns |
| H° | 0.0040 ** | ns | ns | ns | ns | ns | ns | ns | ns |
| CI | 0.0032 ** | ns | ns | ns | ns | ns | ns | ns | ns |
| 3-FQA | ns | ns | 0.0099 ** | 0.0007 *** | ns | ns | 0.0070 ** | 0.0486 * | ns |
| 4-FQA | 0.0003 *** | ns | 0.0076 *** | ns | 0.0056 * | ns | $2.50 \times 10^{-5}$ *** | 0.0003 * | 0.0071 * |
| 5-FQA | 0.0025 ** | ns | 0.0025 ** | ns | 0.0001 ** | ns | 0.0004 ** | 0.0124 | 0.001 |
| 3-CQA | 0.0038 ** | ns | ns | ns | ns | ns | ns | ns | ns |
| 4-CQA | ns | 0.0182 * | ns | ns | ns | ns | ns | ns | ns |
| 5-CQA | 0.0386 * | 0.0031 ** | 0.0379 * | ns | 0.0009 *** | 0.0456 * | ns | ns | 0.0009 *** |
| Total CQAs | 0.0169 * | ns | ns | ns | ns | ns | ns | ns | ns |
| 3,4-diCQA | $1 \times 10^{-5}$ *** | ns | ns | ns | ns | ns | ns | ns | ns |
| 3,5-diCQA | 0.0001 *** | ns | ns | ns | ns | ns | ns | ns | ns |
| 4,5di-CQA | $6 \times 10^{-6}$ *** | ns | ns | ns | ns | ns | ns | ns | ns |
| Trigonelline | $5.1 \times 10^{-5}$ *** | 0.0007 *** | $9.2 \times 10^{-5}$ *** | ns | ns | ns | 0.000 *** | ns | ns |
| Caffeine | ns | ns | ns | ns | ns | ns | ns | ns | ns |
| *p*-coumaric acid | ns | 0.0488 | 0.0340 | ns | 0.0022 * | 0.0022 * | 0.0279 | ns | ns |
| Sucrose | ns | ns | 0.0199* | ns | 0.0021 ** | ns | ns | ns | ns |
| Glucose | $4 \times 10^{-6}$ *** | ns | ns | 0.0075 ** | 0.0479 * | ns | 0 *** | $7.0 \times 10^{-6}$ *** | 0.0437 * |
| Fructose | ns | ns | 0.0007 *** | ns | ns | $7 \times 10^{-5}$ *** | 0.0233 * | ns | 0.0018 ** |
| Arabinose | ns | ns | ns | ns | ns | ns | ns | ns | ns |
| Total sugars | ns | ns | 0.0105 * | ns | 0.005 ** | ns | ns | 0.0293 * | ns |

Symbols represent significant differences at each probability (*p*): ***, $p < 0.001$; **, $p < 0.01$; *, $p < 0.05$; or non-significant differences, ns $p > 0.05$.

In contrast, light conditions affected a much lower number of parameters, such as some chemical compounds (e.g., CQAs, trigonelline), as well as some physical traits (bean mass and apparent density), whereas the altitude vs. light interaction was particularly found in some chemical compounds (FQAs, 5-CQA, trigonelline, *p*-coumaric acid, sucrose and fructose) (Table 1).

### 3.2. Physical Characterization of Green Coffee Beans

The physical attributes of green coffee beans harvested in Gorongosa Mountain were slightly modified by the altitude and shading. Increasing altitude favoured a decrease in temperature and an increase in water availability (greater rainfall, together with frequent fog persistence) (Figures 1 and 2), enabling just a few changes in granulometry, mass and apparent density of green beans.

Most of green coffee beans, regardless of altitude or shade density and full Sun, presented granulometry around 6.70 m, which corresponds to sieve number 17 (Figure 3).

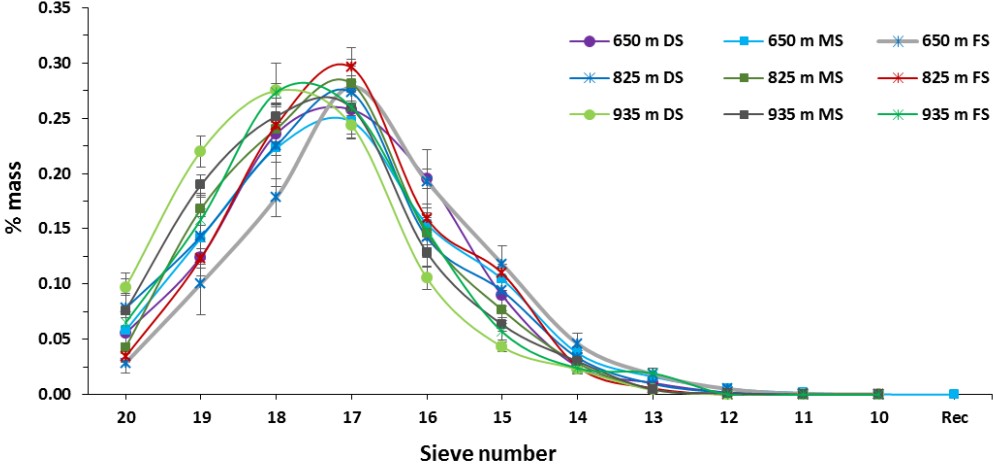

**Figure 3.** Distribution of green coffee beans according to their size, obtained under three light exposure conditions (deep shade, DS; moderate shade, MS; full Sun, FS), and three altitudes (650 m, 825 m and 935 m). Values represent the values ± SE (n = 5).

However, the beans obtained in plants at 935 m of altitude tended to have a greater presence in the sieves 18, 19 and 20, thus showing somewhat larger sizes as compared to those obtained at 650 m and 825 m. On the other hand, a considerable % of beans was found in sieve 17 for 825 m, whereas at 650 m that happened in sieve 16, with some beans found even below sieve 12, thus reflecting a higher proportion of smaller beans at the lowest elevation.

These finding were partly in line with the consistent tendency to greater values of the commercial homogeneity (given by the maximum sum of the percentages of the two most representative sieves) at 935 m and 825 m, whereas the beans from 650 m exhibit values always below 50% (Table 2).

**Table 2.** Variation of the most frequent sieve, average sieve and commercial homogeneity for green coffee beans obtained under three light exposure conditions (deep shade, DS; moderate shade, MS; full Sun, FS), and three altitudes (650 m, 825 m and 935 m).

| Attributes | Light Condition | Altitude | | |
|---|---|---|---|---|
| | | **650 m** | **825 m** | **935 m** |
| **Frequent sieve** | DS | 17.0 ± 0.3 Aa | 17.0 ± 0.5 Aa | 18.0 ± 0.3 Aa |
| | MS | 17.0 ± 0.3 Aa | 17.0 ± 0.3 Aa | 17.0 ± 0.4 Aa |
| | FS | 17.0 ± 0.2 Aa | 17.0 ± 0.2 Aa | 18.0 ± 0.3 Aa |
| **Average sieve** | DS | 17.1 ± 0.0 Aa | 17.1 ± 0.0 Aa | 17.9 ± 0.0 Aa |
| | MS | 16.9 ± 0.0 Aa | 17.3 ± 0.0 Aa | 17.6 ± 0.0 Aa |
| | FS | 16.2 ± 0.0 Aa | 17.1 ± 0.0Aa | 17.5 ± 0.0 Aa |
| **Commercial homogeneity** | DS | 0.49 ± 0.02 Aa | 0.52 ± 0.02 Aa | 0.52 ± 0.01 Aa |
| | MS | 0.46 ± 0.02 Ab | 0.54 ± 0.01 Aa | 0.51 ± 0.02 Aa |
| | FS | 0.45 ± 0.01 Ab | 0.54 ± 0.01 Aa | 0.53 ± 0.02 Aa |

Mean values ± SE (n = 5) followed by the same letter, comparing light conditions within each altitude (A, B) or altitude treatments within each light condition (a, b), are not significantly different according to the Tukey's HSD for mean comparison ($p < 0.05$), always separately for each attribute.

Notably, light conditions promoted only minor (if any) changes in commercial homogeneity at 650 m under MS and FS, that is, under harsher environmental conditions of lower water availability (Figure 1), together with higher temperature (Figure 2).

The mass of 100 coffee beans and apparent density showed maximal values always at the greater altitude, as compared with 650 and/or 825 m (Table 3).

**Table 3.** Mass of 100 green coffee beans (g) and apparent density (g mL$^{-1}$) of green coffee beans obtained under three light exposure conditions (deep shade, DS; moderate shade, MS; full Sun, FS), and three altitudes (650 m, 825 m and 935 m).

| Attributes | Light Condition | Altitude | | |
|---|---|---|---|---|
| | | 650 m | 825 m | 935 m |
| Mass of 100 beans (g) | DS | 16.0 ± 0.5 Aab | 14.3 ± 0.2 Ab | 17.2 ± 0.4 ABa |
| | MS | 15.1 ± 0.2 Ab | 14.5 ± 0.2 Ab | 18.5 ± 0.7 Aa |
| | FS | 14.3 ± 0.4 Ab | 13.2 ± 0.2 Ab | 16.0 ± 0.5 Ba |
| Apparent density (g mL$^{-1}$) | DS | 0.62 ± 0.02 ABb | 0.65 ± 0.02 Aab | 0.69 ± 0.00 Aa |
| | MS | 0.61 ± 0.02 Bb | 0.67 ± 0.01 Aa | 0.69 ± 0.01 Aa |
| | FS | 0.67 ± 0.01 Aab | 0.62 ± 0.01 Ab | 0.69 ± 0.00 Aa |

Mean values ± SE (n = 5) followed by the same letter, comparing light conditions within each altitude (A, B) or altitude treatments within each light condition (a, b), are not significantly different according to the Tukey's HSD for mean comparison ($p < 0.05$), always separately for each attribute.

Also, bean mass showed a tendency to lower values under FS in all altitudes (significantly only at 935 m), whereas apparent density showed significant differences among light conditions only at 650 m, with greater values under FS.

Colour attributes of green beans showed some changes among the altitudes, but were never modified by the light condition, irrespective of the altitude (Table 4). The negative a* values (denoting the strong contribution of the green colour in the bean), and the L* (lightness close to white), were not modified neither by altitude nor by light condition.

**Table 4.** Colour analysis considering colour lightness (L*), the contributions of red or green (a*), and blue or yellow (b*), as well as the calculated parameters of Chroma (C*), Hue angle (H°), and colour index (CI) in grounded green coffee beans from *C. arabica* obtained under three light exposure conditions (deep shade, DS; moderate shade, MS; full Sun, FS), and three altitudes (650 m, 825 m and 935 m) in Gorongosa Mountain.

| Attribute | Light Condition | Altitude | | |
|---|---|---|---|---|
| | | 650 m | 825 m | 935 m |
| L* | DS | 87.7 ± 0.2 Aa | 87.7 ± 0.4 Aa | 88.6 ± 0.3 Aa |
| | MS | 87.7 ± 0.3 Aa | 88.2 ± 0.2 Aa | 88.6 ± 0.6 Aa |
| | FS | 87.7 ± 0.2 Aa | 87.5 ± 0.1 Aa | 88.3 ± 0.2 Aa |
| a* | DS | −3.05 ± 0.06 Aa | −3.05 ± 0.06 Aa | −3.07 ± 0.04 Aa |
| | MS | −2.97 ± 0.10 Aa | −3.16 ± 0.04 Aa | −2.99 ± 0.08 Aa |
| | FS | −3.02 ± 0.06 Aa | −2.92 ± 0.04 Aa | −3.01 ± 0.06 Aa |
| b* | DS | 17.2 ± 0.22 Aa | 16.3 ± 0.1 Ab | 16.2 ± 0.1 Ab |
| | MS | 17.3 ± 0.19 Aa | 16.3 ± 0.2 Ab | 16.2 ± 0.2 Ab |
| | FS | 17.4 ± 0.26 Aa | 16.4 ± 0.2 Ab | 16.3 ± 0.1 Ab |
| C* | DS | 17.7 ± 0.2 Aa | 16.6 ± 0.1 Ab | 16.4 ± 0.1 Ab |
| | MS | 17.5 ± 0.2 Aa | 16.6 ± 0.2 Ab | 16.5 ± 0.3 Ab |
| | FS | 17.7 ± 0.3 Aa | 16.7 ± 0.2 Ab | 16.6 ± 0.1 Ab |
| H° | DS | 99.9 ± 0.2 Aa | 101 ± 0 Aa | 100 ± 0 Aa |
| | MS | 99.7 ± 0.4 Aa | 101 ± 0 Aa | 100 ± 0 Aa |
| | FS | 99.8 ± 0.2 Aa | 100 ± 0 Aa | 100 ± 0 Aa |
| CI | DS | −1.99 ± 0.04 Aa | −2.20 ± 0.04 Ab | −2.14 ± 0.02 Ab |
| | MS | −1.96 ± 0.07 Aa | −2.20 ± 0.04 Ab | −2.08 ± 0.03 Aab |
| | FS | −1.98 ± 0.05 Aa | −2.03 ± 0.03 Aa | −2.09 ± 0.04 Aa |

Mean values ± SE (n = 5) followed by the same letter, comparing light conditions within each altitude (A, B) or altitude treatments within each light condition (a, b), are not significantly different according to the Tukey's HSD for mean comparison ($p < 0.05$), always separately for each attribute.

The positive b* values (reflecting a greater yellow contribution), and C* (measuring light saturation), showed the same pattern of variation, being reduced in the two highest altitudes for all light conditions, as compared with 650 m. In agreement with the yellow colour depicted by b*, both the Hue angle (H°) and colour index (CI) values point to a yellow (/greenish) colour in all treatments, with a slightly greater yellow component at 650 m as compared to the other two altitudes.

### 3.3. Chemical Coffee Green Bean Characterization

Several changes related with altitude and/or light condition were observed in the several studied compounds associated with bean quality, phenolic acids (CQAs, diCQAs, FQAs, $p$-coumaric acid), and nonphenolic compounds (caffeine, trigonelline, sugars) (Table 5).

**Table 5.** Characterization of chemical composition, regarding the contents of trigonelline, caffeine, $p$-coumaric acid, monocaffeoylquinic acids (3-CQA, 4-CQA, 5-CQA), feruloylquinic acids (4-FQA, 5-FQA), and dicaffeoylquinic acids (3,4-diCQA, 3,5-diCQA, 4,5-diCQA) in green coffee beans from *C. arabica* obtained under three light exposure conditions (deep shade, DS; moderate shade, MS; full Sun, FS), and three altitudes (650 m, 825 m and 935 m) in Gorongosa Mountain.

| Atributtes | Light Condition | Altitude | | |
|---|---|---|---|---|
| | | 650 m | 825 m | 935 m |
| Trigonelline (mg g$^{-1}$ DW) | DS | 11.1 ± 0. 3 Aa | 10.4 ± 0.3 Aa | 9.37 ± 1.01 Ca |
| | MS | 11.4 ± 0.6 Ab | 9.91 ± 0.8 Ab | 25.8 ± 0.2 Ba |
| | FS | 11.9 ± 0.5 Ab | 9.63 ± 0.6 Ab | 32.1 ± 1.4 Aa |
| $p$-coumaric acid (mg g$^{-1}$ DW) | DS | 0.41 ± 0.01 Aa | 0.57 ± 0.01 Aa | 0.43 ± 0.07 Aa |
| | MS | 0.43 ± 0.01 Aa | 0.30 ± 0.03 Ba | 0.42 ± 0.04 Aa |
| | FS | 0.41 ± 0.01Aa | 0.42 ± 0.04 ABa | 0.33 ± 0.04 Aa |
| Caffeine (mg g$^{-1}$ DW) | DS | 16.2 ± 0.2 Aa | 13.8 ± 0.4 Aa | 14.7 ± 0.5 Aa |
| | MS | 15.9 ± 0.2 Aa | 14.2 ± 0.2 Aa | 15.1 ± 0.3 Aa |
| | FS | 15.2 ± 0.4 Aa | 13.6 ± 0.3 Aa | 14.8 ± 0.2 Aa |
| 3-CQA (mg g$^{-1}$ DW) | DS | 2.42 ± 0.13 Aa | 2.52 ± 0.26 Aa | 1.92 ± 0.18 Aa |
| | MS | 2.41 ± 0.09 Aa | 2.24 ± 0.19 Aa | 2.35 ± 0.07 Aa |
| | FS | 2.50 ± 0.05 Aa | 2.62 ± 0.07 Aa | 2.17 ± 0.12 Aa |
| 4-CQA (mg g$^{-1}$ DW) | DS | 3.32 ± 0.09 Aab | 3.61 ± 0.11 Aa | 2.67 ± 0.31Ab |
| | MS | 3.51 ± 0.15 Aa | 3.24 ± 0.10 Aa | 3.34 ± 0.11 Aa |
| | FS | 3.60 ± 0.19 Aa | 3.60 ± 0.24 Aa | 3.08 ± 0.14 Aa |
| 5-CQA (mg g$^{-1}$ DW) | DS | 21.6 ± 0.6 Aa | 28.2 ± 1.2 Aa | 25.8 ± 0.8 Aa |
| | MS | 21.8 ± 1.3 Ab | 25.9 ± 0.9 Aab | 32.0 ± 0.8 Aa |
| | FS | 24.9 ± 2.3 Aa | 30.0 ± 1.9 Aa | 28.4 ± 0.9 Aa |
| Total CQAs (mg g$^{-1}$ DW) | DS | 27.4 ± 0.5Aa | 34.3 ± 1.4 Aa | 30.4 ± 3.3 Aa |
| | MS | 27.7 ± 1.4 Ab | 31.3 ± 1.1 Aab | 37.7 ± 1.0 Aa |
| | FS | 31.1 ± 2.7 Aa | 36.2 ± 2.3 Aa | 33.7 ± 1.2 Aa |
| 4-FQA (mg g$^{-1}$ DW) | DS | 0.32 ± 0.04 Aa | 0.16 ± 0.01 Bb | 0.16 ± 0.04 Ab |
| | MS | 0.31 ± 0.02 Aa | 0.18 ± 0.02 ABb | 0.18 ± 0.01 Ab |
| | FS | 0.25 ± 0.02 Aa | 0.23 ± 0.02 Aa | 0.17 ± 0.01 Aa |
| 5-FQA (mg g$^{-1}$ DW) | DS | 2.79 ± 0.16 Aa | 1.84 ± 0.07 Bb | 1.79 ± 0.29 Ab |
| | MS | 2.82 ± 0.09 Aa | 2.13 ± 0.21 ABb | 2.16 ± 0.14 Ab |
| | FS | 2.55 ± 0.14 Aab | 3.14 ± 0.20 Aa | 2.11 ± 0.14 Ab |
| 3,4-diCQA (mg g$^{-1}$ DW) | DS | 0.98 ± 0.08 Aa | 0.65 ± 0.03 Aab | 0.34 ± 0.11 Ab |
| | MS | 1.17 ± 0.10 Aa | 0.44 ± 0.15 Ab | 0.54 ± 0.03 Ab |
| | FS | 1.02 ± 0.09 Aa | 0.56 ± 0.03 Ab | 0.60 ± 0.03 Ab |
| 3,5-diCQA (mg g$^{-1}$ DW) | DS | 3.03 ± 0.10 Aa | 2.04 ± 0.08 Aab | 1.26 ± 0.42 Ab |
| | MS | 3.42 ± 0.28 Aa | 1.49 ± 0.15 Ab | 1.83 ± 0.55 Ab |
| | FS | 2.90 ± 0.29 Aa | 1.76 ± 0.47 Aa | 1.69 ± 0.07Aa |
| 4,5-diCQA (mg g$^{-1}$ DW) | DS | 0.80 ± 0.06 Aa | 0.48 ± 0.04 Aab | 0.37 ± 0.11 Ab |
| | MS | 1.08 ± 0.10 Aa | 0.23 ± 0.03 Ab | 0.53 ± 0.04 Ab |
| | FS | 0.82 ± 0.07 Aa | 0.40 ± 0.10 Ab | 0.45 ± 0.11 Aab |

Mean values ± SE (n = 5) followed by the same letter comparing light conditions within each altitude (A, B, C) or altitude treatments within each light condition (a, b) are not significantly different according to the Tukey's HSD for mean comparison ($p < 0.05$), always separately for each attribute.

Trigonelline content showed one of the most striking responses, since it was markedly modified at the highest altitudes, under MS and FS conditions, showing 126% and 170% increases, respectively, as compared to their respective treatments at 650 m. Furthermore, such increased MS and FS values were also significantly higher than that under DS at 935 m. Notably, no differences were observed between light conditions at 650 and 825 m.

Caffeine and *p*-coumaric acid contents showed mostly small, non-significant, changes, although a reduction of the latter compound was observed under MS and FS at the intermediate elevation.

The contents of the mono CQA isomers, 3-CQA and 4-CQA, showed only minor changes under the studied conditions, with just a minor tendency to lower values at 935 m, and without implications of light conditions in any of the altitudes (Table 4). On the other hand, the most representative CQA (5-CQA) presented consistent tendencies to increase with altitude in all light treatments (but without any difference between light conditions). Maximal values of 5-CQA (as well as of Total CQA) were found under FS at 825 m and MS at 935 m (significantly for the last, as compared to the lowest altitude). Overall, these mono CQA changes modified the proportion of individual CQAs in Total mono CQA content, which in 5-CQA gradually increased from *ca.* 79% to 83% and to 85%, at 650, 825 and 935 m, respectively. Similar values were obtained for all light conditions within each altitude.

The values of FQAs were mostly responsive to altitude (Table 4), with 4-FQA and 5-FQA being reduced at 935 m in comparison with 650 m. Moreover, at 825 m both FQA isomers increased from DS to FS.

As regards the diCQAs, despite some content fluctuations, light conditions did not promote any significant change, regardless of altitude. In sharp contrast, the green bean content of all diCQA isomers (3,4-diCQA, 3,5-diCQA, 4,5-diCQA) greatly declined at the two higher altitudes, frequently to values below half at 935 m, as compared with those found at 650 m.

Soluble sugars did not reveal consistent changes among altitudes and light conditions in the coffee beans (Table 6). Still, under FS sucrose showed greater values at 825 and 935 m (significant for the first). In fact, at 935 m all sugars tended to greater contents under FS, although significantly only for the minor represented glucose and fructose sugars, as compared with DS. In contrast, FS plants showed a tendency to lower values of sucrose, glucose and total sugars at 650 m, Noteworthy is also the fact that sucrose represented the large majority of soluble sugars, between 92 and 95% of their total content.

**Table 6.** Evaluation of soluble sugars in green coffee beans from *C. arabica* obtained under three light exposure conditions (deep shade, DS; moderate shade, MS; full Sun, FS), and three altitude (650 m, 825 m and 935 m) in Gorongosa Mountain.

| Attributes | Light Condition | Altitude | | |
|---|---|---|---|---|
| | | 650 m | 825 m | 935 m |
| Sucrose (mg g$^{-1}$ DW) | DS | 44.3 ± 0.48 Aa | 43.3 ± 2.10 ABa | 43.1 ± 0.53 Aa |
| | MS | 44.6 ± 1.76 Aa | 40.7 ± 1.31 Ba | 42.9 ± 0.84 Aa |
| | FS | 42.9 ± 0.74 Aa | 46.2 ± 0.76 Aa | 44.7 ± 0.65 Aa |
| Glucose (mg g$^{-1}$ DW) | DS | 1.77 ± 0.21 Aa | 1.22 ± 0.07 Ab | 0.66 ± 0.05 Bc |
| | MS | 1.74 ± 0.12 Aa | 1.25 ± 0.09 Ab | 0.84 ± 0.06 ABc |
| | FS | 1.28 ± 0.05 Ba | 0.88 ± 0.10 Ab | 1.05 ± 0.12 Aab |
| Fructose (mg g$^{-1}$ DW) | DS | 0.65 ± 0.05 Aa | 0.96 ± 0.1 Aa | 0.51 ± 0.06 Ba |
| | MS | 0.64 ± 0.03 Aa | 0.76 ± 0.09 Aa | 0.76 ± 0.17 Ba |
| | FS | 0.71 ± 0.03 Ab | 0.69 ± 0.08 Ab | 1.22 ± 0.22 Aa |
| Arabinose (mg g$^{-1}$ DW) | DS | 0.96 ± 0.09 Aa | 0.92 ± 0.09 Aa | 0.92 ± 0.02 Aa |
| | MS | 1.07 ± 0.15 Aa | 0.93 ± 0.10 Aa | 0.93 ± 0.12 Aa |
| | FS | 0.98 ± 0.05 Aa | 0.70 ± 0.08 Aa | 1.00 ± 0.07 Aa |
| Total sugars (mg g$^{-1}$ DW) | DS | 47.7 ± 0.8 Aa | 46.4 ± 2.05 ABa | 45.2 ± 0.50 Aa |
| | MS | 48.1 ± 1.7 Aa | 43.6 ± 1.13 Bb | 45.4 ± 0.88 Aab |
| | FS | 45.9 ± 0.7 Aa | 48.4 ± 0.80 Aa | 48.0 ± 0.51 Aa |

Mean values ± SE (n = 5) followed by the same letter comparing light conditions within each altitude (A, B) or altitude treatments within each light condition (a, b, c) are not significantly different according to the Tukey's HSD for mean comparison ($p < 0.05$), always separately for each attribute.

## 4. Discussion

### 4.1. Coffee Bean Physical Attributes

Altitude driven most of the changes of physical characteristics of coffee bean, whereas only minor impacts were promoted by the irradiance conditions. The apparent density values were close to other field collected beans of *C. arabica* cultivars (*ca.* 0.63–0.68 g mL$^{-1}$) [58,71], similarly to the usual range for the mass of 100 beans (*ca.* 15–20 g) yielded from both Sun or shade management systems [58,72].

Frequent sieve number, average sieve and commercial homogeneity are good indicators for green coffee bean quality, as well as for commercial quality assessment [57]. A frequent sieve around 17 was found for most treatments (Figure 3), in line with the values obtained in *C. arabica* [58]. However, a modest number of smaller beans were found in samples harvested at the lowest altitude, contrasting with larger beans (greater presence in sieves 18 to 20) obtained at 935 m (Figure 1). Also, average sieve, commercial homogeneity (Table 2), bean mass and bean apparent density increased with altitude (Table 2), in line with the reports of maximal bean weight and density at greater altitudes (1400–1500 m) [73], and a rise of the 100 beans weight from 1600–1680 to 1950–2100 m [6]. In our case, the positive altitude impact on these traits might be related with the favourable environment conditions, which were associated with greater water availability (higher rainfall amounts and fog presence, the latter mostly at the greater studied elevation) (Figure 1), together with somewhat lower temperature, particularly in the dry period from May-to-September in MS and FS plants (Figure 2). Lower temperature was pointed to slower the maturation process and to extend the period for bean filling, with beneficial impact on physical (e.g., size and weight) and biochemical (e.g., sugar and acids content) coffee bean attributes [6,10,43,44,74]. In fact, such delayed maturation was evident in Gorongosa Mountain, with a drift of *ca.* two (825 m) and four (935 m) weeks of the harvest date, as compared with 650 m (data not shown). Additionally, although commercial homogeneity showed lower values (<55%) than previously reported for *C. arabica* from Angola [57], only altitude exerts a positive influence in this attribute that significantly increased (at 825 and 935 m) under MS and FS conditions.

Light regime *per se* had no relevant impact on any of the attributes mentioned above when comparing DS and FS conditions (regardless of altitude), in line with unaltered values of yield per plant and weight of 100 fruits in a Cambodia study [52]. Only FS beans showed a systematic trend to lower mass (Table 3), attenuating the bean mass increase due to altitude. Notably, bean mass showed maximal values under the moderate environmental conditions of light (MS) at 935 m. These findings agrees with the absence of impact of shade level in the 100 bean weight, and a trend to lower values under FS regardless of the altitude [6], although this same *C. arabica* cultivar (Costa Rica 95) showed greater bean size under shade [11].

Colour features can also be used to assess bean quality, allowing a fast, reliable, low cost, and non-destructive analysis [24,75], being dependent of (and reflecting), among others, genotype and environmental conditions [47,58]. A few colour attributes were significantly modified by altitude (Table 1), but were irresponsive to light conditions within each elevation level (Table 4). Greater altitude, promoted some decline in b*, C* and CI, mostly independently of the light condition, what was in line with the rise of C*, CI, and L*, and decline of H° at higher temperature (here associated to the lower elevation of 650 m), thus suggesting some degree of altered bean quality [24]. Reflectance measurements indicating greater colour intensity (darker) were associated with a quality decline, due to phenol oxidation, likely with impact on flavour and aroma precursors [75,76]. Additionally, the maintenance of bean quality was related with the stability of colour attributes of blue-green or green tones [75], as it was the case in our samples where the negative a* values reflected a strong contribution of the green colour to the bean in all treatments. In fact, at lower altitude the observed increase in b* just emphasises a moderate increase in the yellow tone, whereas despite the increase in C*, their values were still maintained close to 17 that was reported in non-defective *C. arabica* green beans [77]. In this way, as regards bean

quality, these few changes in colour attributes pointed to a moderate impact of altitude and, and absence of effects of the light regime.

Overall, the studied physical attributes showed that Gorongosa Mountain beans have an appreciable quality, similar to other *C. arabica* cropped genotypes. Yet, a few physical traits (larger beans, with greater mass and density, commercial homogeneity) suggested a quality increase at higher altitude, but these attributes were mostly unresponsive to the light regime.

### 4.2. Chemical Compounds Involved in Bean Quality

Cup quality, namely as regards aroma and flavour, is closely associated with bio-chemical transformations during the roasting process, and depends on the precursors existing in the green bean. Also, the profile of volatile and non-volatile precursors directly related to coffee aroma/flavour formation vary with plant genetic characteristics, geographical and microenvironment factors, agricultural and post-harvest management practices [38,39,78–81]. The contents of several compounds of Gorongosa Mountain beans has fallen within to the ranges previously observed in cropped *C. arabica* genotypes, namely for caffeine and trigonelline, although somewhat below for total mono CQAs [17,24,31,82].

Light conditions (mainly MS and FS) barely affected most of the chemical compounds regardless of the altitude level, in line with the findings that a number of *C. arabica* cultivars responded either positively or neutrally to shaded environments in terms of yield, among them Costa Rica 95 [30]. In contrast, changes in the chemical attributes were promoted mainly by altitude (Table 1). Among the studied compounds, trigonelline showed one of the more striking changes with altitude, with rises to *ca.* 2.3 (MS) and 2.7 (FS) fold at 935 m when compared to their respective values at 650 m (or at 825 m) (Table 5). Greater trigonelline value, together with more intense aroma, was also found in lower canopy beans, what was attributed to their maturation under better microenvironment conditions [79], although contrasting with the nearly stable values from 900 to 1450 m reported by [78]. This compound is a known precursor of volatile compounds that contribute to the aroma and taste after roasting [45,82]. In fact, improved cup quality can be sensed with relatively small increases from 9.6 to 13.4 mg g$^{-1}$ DW in the green bean [39], suggesting that the bean harvested at 935 m would have a significant increase in cup aroma and taste in MS and FS (but not in DS) plants. This is in line with lower trigonelline values under shade for this genotype [11], as well as with a greater quality for specialty coffee observed under high altitude combined with full Sun or medium shade level, including greater quality scores for total specialty cup quality, overall cup preference, acidity, body, flavour, and after taste [6].

Caffeine is a very important compound to flavour and is also a bioactive element [36]. Their patterns of accumulation are highly variable, and have been found to slightly increase with higher elevation above 1200 m [78], to decrease at 1400–1500 m [73] or 1950–2100 m with moderate shade of 40–55% [6], or to remain unaffected by altitude or shading [83]. Caffeine and polyphenol *p*-coumaric acid, together with CGAs, contribute to coffee bitterness, and their rise under higher temperature conditions was pointed to result in poor bean quality [2,44]. That was not the case in our study, since the hydroxycinnamic *p*-coumaric acid and caffeine showed only marginal changes regardless of altitude or light, with the latter being (together with arabinose) the most stable compound, without significant variance changes under the single and combined altitude and light conditions (Tables 1 and 5).

The mono CQAs isomers are the most represented subgroup of CGAs, with strong influence in the development of coffee cup flavour [55,56,83]. In our case, mono CQAs showed an absence of changes with light conditions, and some divergent trends with altitude, depending on the isomer. The 3-CQA and 4-CQA consistently tended to decline at 935 m (significantly only for 4-CQA under DS), whereas 5-CQA (and the total of mono CQAs) increased at 825/935 m as compared with 650 m (significantly only under MS at the highest elevation) (Table 4). Additionally, 5-CQA, which is by far the largest fraction of total CGAs, increased its proportion in the total mono CQA content in all light conditions. This result agrees with the decline of 5-CQA promoted by supra-optimal temperatures [24], here

associated with the lower values at the lowest altitude. Coffee berries grown at high altitude experience lower temperatures, conditions that increase the period of fruit maturation, as observed also in our case. Such extended period can allow a higher accumulation of phenolic compounds, among them chlorogenic acids. Still, there are divergent reports regarding the impact of elevation on mono CGAs values, which can tend to lower values (1950–2100 m vs. 1600–1680 m) [6], or to rise [76,81] at greater altitudes, particularly 3-CQA and 5-CQA. In fact, the 5-CQA rise has been associated with first grade coffee beans, both under shaded and unshaded conditions [83], since this compound has been characterized as low acidic, with a small amount of bitterness [37]. Additionally, 5-CQA in stored green coffee was also pointed as responsible for sensory perception of coffee freshness [84], but a certain degree of uncertainty still remains regarding their impact on the sensory properties of expresso coffee beverage. As highlighted in a recent review [37], 5-CQA might not contribute substantially to sensory beverage quality by itself, since its large enzymatic degradation (98%) in brewed coffee did not significantly alter the beverage flavour [85]. Degrading chlorogenic acid lactones to CGAs induced a decline in the bitterness attributed to lactones, although that did not promote other sensory attributes. Yet, when 5-CQA was hydrolyzed to caffeic acid, the drink was found to be more sour, bitter, and burnt [85,86]. Hence, CGAs seemed to have a limited direct contribute to coffee flavour after roasting, while their derivatives did [37].

As for mono CQAs, also the FQAs were moderately impacted by the light, except at 825 m, where beans collected from FS plants showed greater 4-FQA and 5-FQA contents (Table 5), in line with the significantly higher values of FQAs present in unshaded than in shaded coffee bean samples [83]. On the other hand, the impact of altitude was clearer with both compounds showing declines at 825 and 935 m, corroborating earlier findings of greater FQA values in beans from lower altitude [83]. Such lower FQAs values at 935 m (and in DS and MS at 850 m) could improve the sensory quality of this coffee, since high FQA contents were reported in coffees with poor cupping scores after roasting [39]. However, it seems likely that the FQAs specific negative/positive role on the beverage will be largely dependent of some interactions with other chemical components of each specific coffee [37].

As for most compounds previously discussed, all diCQA isomers (3,4-diCQA, 4,5-diCQA and 3,5-diCQA) did not show any significant change with the light conditions within each altitude, but where clearly reduced at the two higher altitudes. Similar absence of shade effect on diCQAs, as well as lower 3,4-diCQA and 4,5-diCQA contents at high than at low altitude were previously reported [83]. Such, diCQAs reduction at higher altitudes might benefit cup quality, since diCQA mixtures were suggested to present a greater bitterness, together with a metallic taste and astringency [35,87]. Nonetheless, much remains to be elucidated due to contradictory reports, since 3,4-diCQA was associated with sweetness and full body, and 3,5-diCQA with astringency and immature bean taste [88], but 3,5-diCQA and 4,5-diCQA contents were also positively associated with higher cup quality scores [89].

Sugars are another group of important compounds, which undergo complex changes during the roasting that contribute to the cup organoleptic profile [90]. For instance, sucrose degradation during roasting plays a major role in flavour formation through Maillard reactions, although its relation with sensory characteristics remains unclear in some cases [56]. As for CGAs, an extended period of fruit maturation due to lower temperature/higher altitude was pointed to increase coffee bean sugars and lipids [6,43,55], but greater sucrose content of coffee beans from low than at middle and high altitudes was also reported [83]. In our case, sucrose was by far the most represented soluble sugar irrespective of the environmental conditions (Table 6), but did not show a consistent pattern of variation with altitude. That holds true for all sugars except glucose, which declined from 650 to 935 m, although without impact on total sugars. However, it seems noteworthy that FS management tended to somewhat lower sucrose, glucose and total sugars contents at the hotter conditions (650 m), contrasting with the greater absolute values for all sugars at the

coolest 935 m (significantly for glucose and fructose), together with the above mentioned greater content of trigonelline, always as compared with MS and DS.

## 5. Conclusions

The present study highlights the impact of light conditions (full Sun and agroforestry system managements) and altitude (associated with temperature and water availability) on the physical and chemical attributes of *C. arabica* green coffee beans in Gorongosa Mountain.

Light conditions (mainly MS and FS) barely affected most of the studied physical and chemical traits of the coffee bean irrespective of elevation, as also found in other works in a number of *C. arabica* cultivars. Among the few physical traits exceptions, the bean mass consistently tended to lower values under FS in the three altitudes (significantly only as compared with MS at 935 m), and the density was greater under FS at 650 m, whereas colour attributes were irresponsive to light conditions. As regards the chemical compound contents, significant impacts were observed sporadically. An improved sensory profile might be expected (after roasting) in the beans regarding the rises of trigonelline (FS and MS at 935 m), all sugars (FS at 935 m), and *p*-coumaric acid decline (MS and FS at 825 m). On the other hand, the FQAs rise (FS at 825 m) could have a negative impact, thus adding a greater uncertainty regarding potential quality modifications of the yielded bean.

Contrasting with light conditions, altitude promoted changes in a large number of attributes with potential implications to bean quality. Altitude extended the fruit maturation period by four weeks from 650 to 935 m. Among the physical attributes, average sieve (consistent tendency), bean commercial homogeneity, mass, and density significantly increased at 935 m, whereas the beans became less yellowish and brighter at 825 and 935 m (b*, C*), usually as compared with the beans from 650 m elevation, all of which pointing to good bean trade quality. Additionally, at 935 m increases of trigonelline and 5-CQA (both under MS and FS) were observed, whereas FQAs and diCQAs isomers declined (regardless of the light condition), all of which likely contributing to improve the sensory cup quality at the highest altitude. Notably, that could also be the case for the FS plants that showed greater sugars content (and trigonelline) at the coolest 935 m, in sharp contrast with the trend to the lowest sugar values under the warmer conditions at 650 m, as compared to their MS and DS counterparts at the same elevation. Other major compounds (*p*-coumaric acid and caffeine), showed mostly inconsistent variations, with the latter presenting a great stability with an absence of significant changes regardless of the of single and combined of altitude and light.

Overall, light (full Sun and shade under AFS) did not greatly and consistently modify the bean quality physical and chemical attributes. In contrast, altitude, likely associated with lower temperature, greater water availability through rainfall and fog, and extended fruit maturation period, was a major driver for bean those attribute modifications, ultimately improving coffee green bean quality.

**Author Contributions:** Conceptualization, F.L.P., A.I.R.-B. and J.C.R.; Formal analysis, C.T.C., A.V.J.M., A.E.L., I.P.P., R.M., C.C., R.C., F.O.R., I.M., P.S.-C., F.C.L., A.I.R.-B. and J.C.R.; Funding acquisition, A.I.R.-B. and J.C.R.; Investigation, C.T.C., A.V.J.M., A.E.L., I.P.P., R.M., C.C., R.C., F.O.R., I.M., F.C.L., F.L.P., A.I.R.-B. and J.C.R.; Methodology, A.E.L., I.P.P., R.M., C.C., P.S.-C. and J.C.R.; Project administration, A.I.R.-B. and J.C.R.; Supervision, A.E.L., R.C., F.C.L., F.L.P., A.I.R.-B. and J.C.R.; Visualization, C.T.C., A.V.J.M. and A.E.L.; Writing—original draft, C.T.C., F.O.R., I.M. and J.C.R.; Writing—review & editing, C.T.C., A.V.J.M., A.E.L., I.P.P., R.M., C.C., R.C., F.O.R., P.S.-C., F.C.L., F.L.P., A.I.R.-B. and J.C.R. All authors have read and agreed to the published version of the manuscript.

**Funding:** This work was supported by national funds of Camões—Instituto da Cooperação e da Língua (Portugal), Agência Brasileira de Cooperação (Brazil), and Parque Nacional da Gorongosa (Mozambique), through the project GorongosaCafé (TriCafé). Portuguese national funding support was also provided by Fundação para a Ciência e a Tecnologia, I.P. (FCT), through the grant SFRH/BD/135357/2017 (C. Cassamo), through the Scientific Employment Stimulus—Individual Call (CEEC Individual)—2021.01107.CEECIND/CP1689/CT0001 (IM) and to the research units

UIDB/00239/2020 (CEF) and UIDP/04035/2020 (GeoBioTec). Grant from CNPq, Brazil, to F. Partelli, is also greatly acknowledged.

**Data Availability Statement:** Data is contained within the article.

**Acknowledgments:** The authors would like to Tech. Paula Alves for laboratory assistance, and to Jossefo Saliva (GNP) for fruit harvesting.

**Conflicts of Interest:** The authors declare no conflict of interest.

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
