# Peer review of "Shade and Altitude Implications on the Physical and Chemical Attributes of Green Coffee Beans from Gorongosa Mountain, Mozambique"

_agronomy, doi:10.3390/agronomy12102540_

Round 1
Reviewer 1 Report
The article Shade and altitude implications on the physical and chemical attributes of green coffee beans from Gorongosa Mountain, Mozambique brings important information regarding beans guality.
The introduction is well written, but clear hypotheses are missing.
Material and methods address different aspect of the research. Somewhat weak is the description of environmental conditions, especially radiation regime. The values of full sun radiation are usually 2000 µmol/m2s, so please describe how your values were obtained (where was the measuring point). The same hold true for deep shade, where radiation was rather high. Your value is more than 5% of full-light which is rather high for understory.
May be better to use full sun, half(semi)-shade, shade.
The same hold true for temperatures. Did you tested the differences? Please give averages and SDs.
What about overall production? Do you have any data on beans production per area? This is probably also important.
Please give full names of atributes in the tables 3 and 4.
Discusion is quite long. Please shorten it by 30% and make it more cosistant. For example the first paragraph is the repetition of introduction.
Skip statement with reference from Conclusions.
Author Response
Comments and Suggestions for Authors
The article Shade and altitude implications on the physical and chemical attributes of green coffee beans from Gorongosa Mountain, Mozambique brings important information regarding beans guality.
Answer:
We wish to thank the reviewer for his/her time spent and for the positive comments, suggestions, and corrections. We introduced the suggested modifications, but when needed we provide a more complete answer regarding some specific questions.
The introduction is well written, but clear hypotheses are missing.
Answer:
We (as in many published papers) do not state a hypothesis. Instead, our option is to state the subject/aim of what we are studying and how we propose to achieve our goals, what in our view is more informative to the reader about what we deal with and how we do it. We modified and added some text in order to become clearer.
Material and methods address different aspect of the research. Somewhat weak is the description of environmental conditions, especially radiation regime. The values of full sun radiation are usually 2000 µmol/m2s, so please describe how your values were obtained (where was the measuring point). The same hold true for deep shade, where radiation was rather high. Your value is more than 5% of full-light which is rather high for understory.
Answer:
The reviewer is correct when pointing this value. However that value represents the maximum of irradiance (2000 µmol m-2 s-1). Yet, as stated, the provided irradiance values are a diurnal average, from values obtained 4 times a day (9-10 h; 11-12 h; 13-14 h; 15-16 h) along 6 months (April, May, June, July and September), using the light sensor from an infrared gas analyzer (IRGA) in clear sky days. We modified the text to better explain this.
Furthermore, the irradiation value for shade treatments are not high considering that shade trees are tall and with their leaves mostly in the upper part of the canopy (and not in a dense canopy), thus letting diffusion light enter beneath their canopy . Still these were the values measured, having also in mind that we choose the conditions that could be used with coffee but also to the restoration of the degraded rainforest of Gorongosa mountain (as stated almost at the end of the Introduction section).
May be better to use full sun, half(semi)-shade, shade.
Answer:
We thank the reviewer for the suggestion but we prefer to maintain our previous designation since we believe that better characterize the light conditions, namely as regards the fact that the most shaded treatments is really a deep shade (DS) situation, whereas the intermediate light level configures a moderate shade situation (as several authors claim that productivity might not be affected until 60% of shade, depending on genotype – please see the recent work Koutouleas et al. 2022 included in the Reference list)
The same hold true for temperatures. Did you tested the differences? Please give averages and SDs.
Answer:
The given temperatures (Fig. 2) result from the data taken at each site as was state at the end of 2.1 of Material and Methods. “with Temperature External Data Logger devices (ONSET, Pro v2 Logger U23-00x), installed in each of the three altitudes and three light conditions”. As better stated now these monthly average values resulted from: 1) 96 measurements per day (thus, taken every 15 min). These 96 values per day were used to obtain a daily mean. Monthly averages in each date were then calculated using the 30 daily values. As in all the other figures and tables the measure of sample variation was not given as SD but as SEs (mean ± SE). Therefore, due to the large data sample the SE bars are embedded in these Fig 2 graphs, that is, they are in the graphs but in some cases they are not visible due to the very large number of data points that contributed to the presented means.
We did not tested temperature differences.
What about overall production? Do you have any data on beans production per area? This is probably also important.
Answer:
Yield is an indisputable very important issue, but our manuscript is focused on quality issues for the already established cropped systems.
Please give full names of atributes in the tables 3 and 4.
Answer:
We do not understand this comment. Table 3 is about colour attributes. Each parameter is given in an abbreviated form in the table, but is fully identified in the legend (as it should be). Table 4 regards chemical compounds, which in most cases are also given in an abbreviated form (e.g., 3-CQA, 4-FQA, etc.). However, these compounds are also properly identified in the legend, similarly to what happens in many other papers. In this way, we believe that the tables are quite clear as they were previously submitted but we introduced some text in the legends when needed.
Discusion is quite long. Please shorten it by 30% and make it more cosistant. For example the first paragraph is the repetition of introduction.
Answer:
As suggested, the first paragraph of the Discussion was removed (and integrated in the Introduction). Still, in our view the Discussion size (about 6 pages with 1.5 line spacing and Times New Roman 12 size) is adequate for the large number of attributes included in this manuscript, together with the study of the single and combined effect of altitude and light. However, we did our best to comply with the request to reduce the Discussion, by performing a complete revision of this section, although trying to avoid to impoverish the manuscript. We hope that the reviewer understands our point of view (and our efforts in the Discussion reduction) and consider the new Discussion as having a good size (and content) which has now about 4,5 pages.
Skip statement with reference from Conclusions.
Answer:
It was altered as requested.
Reviewer 2 Report
The manuscript deals with the study of effect of altitude and shade on composition of beans and morphopysical traits of grean coffee under natural conditions of Gorrongoza mountain.
Authors have mearused biochemical and morphophysical characteristics of coffee beans. The used methods are sounds. The manuscript is welle written.
Nevertheless, there are several concerns that limit the soundness of the manuscript
1-the exeprimental design is not presented
2- the statistical analyses are not clear. Indeed, authors cite (L326-327) "data were analysed using a two-way ANOVA, considering the effects altitude and/or light conditions". It is not clear and is related to the experimental design not presented. AS authors prensent two factor (shade and altitude) is is more judicious to use a split plot design. therefore, analyses of variance should be redone. Moreover, authors did not discuss the interaction between the two factors.
3 authors discuss, physiologically the observed results based on the impact of temperature and water. They did not consider the UV impact on both growth and quality of beans. This should be considered since the UV act negatvely on phytohormones and inhibit the fruits development.
Author Response
Comments and Suggestions for Authors
The manuscript deals with the study of effect of altitude and shade on composition of beans and morphopysical traits of grean coffee under natural conditions of Gorrongoza mountain.
Authors have mearused biochemical and morphophysical characteristics of coffee beans. The used methods are sounds. The manuscript is welle written.
Nevertheless, there are several concerns that limit the soundness of the manuscript
Answer:
We wish to thank the reviewer for his/her time spent and for the positive comments, suggestions, and corrections. We introduced the suggested modifications, but when needed we provide a more complete answer regarding some specific questions.
1-the exeprimental design is not presented
Answer:
The experimental design was extensively reviewed, with the introduction of new text in the Material and Methods section (at 2.1 sub-point) to comply and overcome this comment.
2- the statistical analyses are not clear. Indeed, authors cite (L326-327) "data were analysed using a two-way ANOVA, considering the effects altitude and/or light conditions". It is not clear and is related to the experimental design not presented. AS authors prensent two factor (shade and altitude) is is more judicious to use a split plot design. therefore, analyses of variance should be redone. Moreover, authors did not discuss the interaction between the two factors.
Answer:
Although we consider that the statistical analysis was properly applied, we also recognize that a split plot design could be more consensual, considering each altitude as the main plot and the three levels of irradiance as the sub-plot. The ANOVA was done again taking this into account (the ANOVA results are now provided in Table 1), although without any relevant impact in the mean comparisons shown in Figures and tables).
Discussion
3 authors discuss, physiologically the observed results based on the impact of temperature and water. They did not consider the UV impact on both growth and quality of beans. This should be considered since the UV act negatively on phytohormones and inhibit the fruits development.
Answer:
We thank the reviewer for this comment, although we did not share his/her view. In fact, the UV question might be an interesting issue, but is quite aside of the focus of this paper, and we did not measured UV levels. When the reviewer says that, “UV act negatively on phytohormones and inhibit the fruits development” that would lead to an absence of coffee crop under full sun exposure worldwide in comparison with shaded systems. That is not the case. In fact, many countries use this kind of full Sun management to obtain greater yields than under shade (where a lower UV is present), although with greater need for fertilization and water availability. That is the case of Brazil, the largest coffee world producer, where the large majority of this crop is managed under full Sun conditions. Furthermore, at higher altitudes, greater UV would be found, and yet several works (see reference list and discussion) claim that a better coffee bean quality is usually found at higher elevations. This is usually attributed to the lower temperature (allowing an extended fruit maturation period) and higher water availability (crucial to plant and fruit development), but not to UV. Even in our results, the bean produced under full Sun conditions at 935 m have some features of quality (e.g., a tendency to greater sugar content) greater than those with moderate shade (MS) or deep shade (SD).
Still, we tried to found some work(s) relating UV exposure to altered coffee fruit/bean production or quality, but we were not able to find a work on that subject.
Reviewer 3 Report
Please see the attached file.

Author Response
Comments and Suggestions for Authors
Please see the attached file.
Authors tried to use the natural condition to figure out the difference between growing environmental factors and showed the physical and chemical differences of green coffee beans. Authors tried to use the natural condition to figure out the difference between growing environmental factors and showed the physical and chemical differences between green coffee beans. However, some flaws could be improved to increase the readability of this manuscript.
Answer:
We wish to thank the reviewer for his/her time spent and for the positive comments, suggestions, and corrections. We introduced the suggested modifications, but when needed we provide a more complete answer regarding some specific questions.
- The reference does not match the requested format.
Answer:
Thanks for this comment. We introduced the required change at this time, that is, when the Reference list is stabilized/final.
- The first paragraph is too long to read. Please consider shortening and putting the emphasis on the close related information to the topic.
Answer:
We reviewed, divided and shortened the paragraph.
- It is hard to get enough information from the table data. The tables of statistical data by two-way
The suggested table format may show like below:
|
|
|
Mass of beans |
Density |
… |
|
Light |
DS |
|
|
|
|
|
MS |
|
|
|
|
|
FS |
|
|
|
|
Altitude |
650 m |
|
|
|
|
|
825 m |
|
|
|
|
|
935 m |
|
|
|
|
Light |
|
*, **, ns |
|
|
|
Altitude |
|
|
|
|
|
Interaction |
|
*, **, ns |
|
|
Answer:
We provide now a table with the ANOVA results, whereas statistical indexes for mean comparison (after a Tukey HSD test, as described in the Material and Methods) are found in the tables, close to the values. This is the most accurate way of providing these results whose mean differences are discussed (based on the mean comparison results from the mentioned Tukey HSD test). Additionally, we also clarified the experimental design.
Reviewer 4 Report
The presented article contains the results of the influence of shade and altitude implications on the physic-chemical characterization of green coffee beans from Gorongosa Mountain. This paper is well written in general. The experiment was well planned and the data are properly analyzed and deeply discussed. I have only a few comments.
The citation should be given in numbers according to the Agronomy journal.
Lines 2267-274 Authors should write the equations properly in separate lines (using a Microsoft equation tool) and not directly in the text.
References should be harmonized with Agronomy journal standards.
Author Response
Comments and Suggestions for Authors
The presented article contains the results of the influence of shade and altitude implications on the physic-chemical characterization of green coffee beans from Gorongosa Mountain. This paper is well written in general. The experiment was well planned and the data are properly analyzed and deeply discussed. I have only a few comments.
Answer:
We wish to thank the reviewer for his/her time spent and for the positive words, comments, suggestions, and corrections. We introduced the suggested modifications, but when needed we provide a more complete answer regarding some specific questions.
The citation should be given in numbers according to the Agronomy journal.
Answer:
Thanks for this comment. We introduced the required change at this time, that is, when the Reference list is already stabilized/final.
Lines 267-274 Authors should write the equations properly in separate lines (using a Microsoft equation tool) and not directly in the text.
Answer:
If the reviewer is referring the equations for colour analysis (point 2.3.2), we were not able to perform the requested change. For that our apologies. However, we believe that the formulae are quite clear as they are. Nevertheless, if the journal imposes such kind of presentation we believe that the editorial system can provide such change in a final (/proofs) version.
References should be harmonized with Agronomy journal standards.
Answer:
Thanks for this comment. References were provided consistently along the reference list. We believe that, if needed, we will ask the editorial services of Agronomy to introduce such changes in the approved version of the manuscript (as occurred in previous publications of these authors in MDPI journals)
Round 2
Reviewer 1 Report
The article is significantly improved. There is still a mistake in Fig. 3, where lable indicate %, but the numbers along the axis presumably show a ratio. This shoud be corrected.
Author Response
Comments and Suggestions for Authors
The article is significantly improved. There is still a mistake in Fig. 3, where lable indicate %, but the numbers along the axis presumably show a ratio. This shoud be corrected.
Answer:
We thank the reviewer for this comment. However, the figure is correct. In the YY axis is displayed the percentage (%) of bean mass retained in each sieve (meaning that the greater % of mass retained in higher size sieves, the greater the size of the beans). That is also stated in the 2.3.1 sub-point when it says that “the weight retained in each sieve was converted to percentage of 100 g of bean” (in this sense is a ratio) as described in Kath et al. 2021, after ISO 2011 (both of which are in the list of references of our manuscript [60 and 61]).
Reviewer 2 Report
I am satisfied with the changes made in this new version of the manuscript. This manuscript has been significantly improved.
Author Response
Comments and Suggestions for Authors
I am satisfied with the changes made in this new version of the manuscript. This manuscript has been significantly improved.
Answer:
We thank the reviewer for this comment and acceptance of this new version of the manuscript
Reviewer 3 Report
The revised version has been modified for major flaws and is suggested to be accepted.
Author Response
Comments and Suggestions for Authors
The revised version has been modified for major flaws and is suggested to be accepted.
.
Answer:
We thank the reviewer for this comment and acceptance of this new version of the manuscript